# Anomalous diffusion along metal/ceramic interfaces

Aakash Kumar [1], Hagit Barda[2], Leonid Klinger[2], Michael W. Finnis[3], Vincenzo Lordi[4], Eugen Rabkin[2] & David J. Srolovitz [1,5,6]

Interface diffusion along a metal/ceramic interface present in numerous energy and electronic devices can critically affect their performance and stability. Hole formation in a polycrystalline Ni film on an $\alpha$-Al$_2$O$_3$ substrate coupled with a continuum diffusion analysis demonstrates that Ni diffusion along the Ni/$\alpha$-Al$_2$O$_3$ interface is surprisingly fast. Ab initio calculations demonstrate that both Ni vacancy formation and migration energies at the coherent Ni/$\alpha$-Al$_2$O$_3$ interface are much smaller than in bulk Ni, suggesting that the activation energy for diffusion along coherent Ni/$\alpha$-Al$_2$O$_3$ interfaces is comparable to that along (incoherent/high angle) grain boundaries. Based on these results, we develop a simple model for diffusion along metal/ceramic interfaces, apply it to a wide range of metal/ceramic systems and validate it with several ab initio calculations. These results suggest that fast metal diffusion along metal/ceramic interfaces should be common, but is not universal.

[1] Department of Materials Science and Engineering, University of Pennsylvania, Philadelphia, PA 19104, USA. [2] Department of Materials Science and Engineering, Technion - Israel Institute of Technology, 3200003 Haifa, Israel. [3] Thomas Young Centre, Department of Materials and Department of Physics, Imperial College, London SW7 2AZ, UK. [4] Materials Science Division, Lawrence Livermore National Laboratory, Livermore, CA 94550, USA. [5] Department of Mechanical Engineering and Applied Mechanics, University of Pennsylvania, Philadelphia, PA 19104, USA. [6] Department of Materials Science and Engineering, City University of Hong Kong, Kowloon, Hong Kong SAR. These authors contributed equally: Aakash Kumar, Hagit Barda  Correspondence and requests for materials should be addressed to A.K. (email: aakashk@seas.upenn.edu) or to E.R. (email: erabkin@technion.ac.il) or to D.J.S. (email: srol@seas.upenn.edu)

Metal/ceramic interfaces are ubiquitous building blocks for a wide range of technologies, from semiconductor devices (metal/gate-oxides)[1] to thermal barrier coatings in gas-turbines (metal/Yttria Stabilized Zirconia (YSZ))[2] to all-solid-state batteries (Li anode/electrolyte)[3]. Device performance thus directly depends on the integrity of these metal/ceramic interfaces (often limited by atomic transport along the interface). For example, it was shown that the kinetics of electromigration-induced Cu precipitation and dissolution in Al-Cu bamboo-structure interconnects deposited on a TiN barrier layer is controlled by Cu diffusion along the metal/TiN interface[4]. The morphological evolution of thin passivated metal films is also commonly controlled by metal diffusion along the metal-passivation layer interface (i.e., $Cu/SiO_2$)[5,6]. These examples demonstrate that rational device design demands an improved understanding of transport along metal/ceramic interfaces[7]. While an extensive literature exists on the mechanical strength[8], atomic structure[9], and chemical composition[10] of metal/ceramic interfaces, little is known about atomic diffusion along this channel. Diffusion along extended crystal defects (e.g., surfaces, dislocations, and grain boundaries (GBs)) is commonly much more rapid than bulk diffusion[11,12]. The widely-quoted hierarchy of diffusivities is $D_{bulk} \leq D_{dislocation} \leq D_{GB} \leq D_{surface}$[13,14]. Here, we focus on where metal/ceramic interfaces fall within this hierarchy, to discover the features that control such diffusion, and use these to predict its magnitude.

Ideally, bicrystal samples should be used to directly measure interface diffusivities; a diffusant is deposited on a bicrystal surface, the sample is annealed, and the composition profile is measured, leading to the determination of the interface diffusion coefficient. Only a very small set of measurements for metal/metal[15] and metal/semiconductor[16] interfaces have been made. We are unaware of any such direct measurements of metal diffusion along metal/ceramic interfaces. However, there is indirect evidence to suggest that metal/ceramic interfaces may be high-diffusivity paths for metal atoms. For example, Gan et al.[17] showed that Cu diffusion along the (SiN,SiC)/Cu interface is faster than bulk diffusion in Cu. They extracted the interface diffusivities from the kinetics of stress relaxation in the Cu films, making their results indirect and strongly model-dependent. Arnaud et al.[18] demonstrated that Cu diffusion along a Cu/SiN interface plays a prominent role in electromigration and estimated the activation energy of this interface diffusion. Recently, indirect evidence of fast diffusion along metal/ceramic interfaces was found in the solid-state dewetting of metal films on oxide substrates[19,20]. It is reasonable to conjecture that diffusion along metal/ceramic interfaces will be comparable to that along other internal interfaces (e.g., GBs in metals). Yet, metal–oxygen bonds (in oxide ceramics) may be much stronger than those between metal atoms, suggesting rather that diffusion along metal/ceramic interfaces may be suppressed (relative to GBs in metals). Fast diffusion along GBs is usually attributed to their lower density or higher free volume (relative to grain interiors); a concept supported by the observation that diffusion along coherent twin boundaries appears no faster than in the bulk[21]. In fact, Chen et al.[22] showed that coherent twin boundaries slowed electromigration in Cu. Nonetheless, there is evidence that diffusion along coherent (and semicoherent) metal/ceramic interfaces is rapid[23,24]. This seems contrary to the association of fast transport with low atomic density.

In this work, we report experimental evidence of fast diffusion along the $Ni/\alpha$-$Al_2O_3$ (sapphire) interface. We explore the origin of this effect as a function of environment ($O_2$ partial pressure) using ab initio calculations of point defects within these materials and near their interface. We establish that in this case, the dominant defects for interface transport are Ni vacancies and that the diffusivity of Ni along the interface is anomalously fast. We then present a simple model that generalizes these results to a very wide range of metal/ceramic interface systems and validate the model through examination of additional metal/ceramic interface systems.

## Results

**Experimental evidence of fast interface diffusion.** We examine the case of an annealed 45 nm thick Ni film deposited on the (0001) surface of sapphire. All Ni grains have $\langle 111 \rangle$ surface normal with two in-plane orientations (rotated by 60° about the normal with respect to one another); this is referred to as a maze microstructure. We observed that the grain surfaces are flat, except for the presence of ridges and/or grooves at some GBs and holes within some of the grains[25] (see Fig. 1a).

The presence of isolated holes, not connected to GBs, is a clear indication that isolated/embedded grains (see Fig. 1a) sink and disappear, leaving through-thickness holes. These holes form in an early stage of solid-state dewetting. The hole indicated by the turquoise-colored arrow in Fig. 1a is surrounded by a slightly elevated ridge (see Fig. 1b, c). Integrating the profile around this hole from the AFM topography reveals that the volume contained in the ridge is ~$0.5 \times 10^{-2}$ $\mu m^3$, while the volume of the material removed to form the hole is ~2X larger. The "sunken" grain indicated by the magenta arrow in Fig. 1a shows no such ridges (Fig. 1d, e). In the classic theory of GB grooving[26], all of the material removed around the GB is transported away from the GB into the ridge by surface diffusion. We now consider where the missing Ni atoms go and how these holes form.

We suggest that the missing Ni in the grooving/hole formation, diffuses down the GB and then along the Ni/sapphire interface (see Fig. 2a). The material diffusing along the GB is supplied by surface diffusion toward the GB (i.e., rather than away from the GB as in classical GB grooving[26]). For this process to continue, Ni diffusion along the Ni/sapphire interface must be rapid. In this case, the film surrounding the sunken grain must thicken by the accretion of Ni atoms at the Ni/sapphire interface[25]. Similar homogeneous thickening of a metal film associated with material redistribution along the metal/ceramic interface was recently reported in Al/sapphire[23].

We test this hypothesis by developing a short-circuit diffusion model (Supplementary Method 1) that describes the surface topography evolution of a thin metal film on a ceramic substrate via simultaneous surface, GB, and interface metal diffusion (see Fig. 2a). Numerical solution of the evolution of the surface profile for the case of a small, axisymmetric grain of radius $R_0$ (Supplementary Table 1, Supplementary Method 1) embedded in a continuous film is shown in Fig. 2b for three different interface diffusivities at the time when the GB groove hits the substrate (i.e., hole nucleation and onset of solid-state dewetting). This figure clearly shows that interface diffusion greatly enhances the rate of hole formation and reduces the amplitude of the elevated rim demonstrating that our model is consistent with the observations.

We analyze the experimental case of Fig. 1b, c (where a small ridge forms around the sinking grain) employing reasonable values of GB and surface diffusivities for Ni via our model in order to determine the interface diffusivity required to explain the "missing" Ni (the difference between the Ni forming the ridge and that from the sinking grain, $\Delta V = 0.5 \times 10^{-2}$ $\mu m^3$ after a 10 min annealing at 700 °C). These experimental observations yield $D_i \approx D_{gb}$ (for a random large angle GB in Ni[27]) as shown in Fig. 2c. We note that given the variability of GB and surface diffusivity with bicrystallography and the uncertainty in the experimental measurements, we consider this to be an order-of-magnitude estimate.

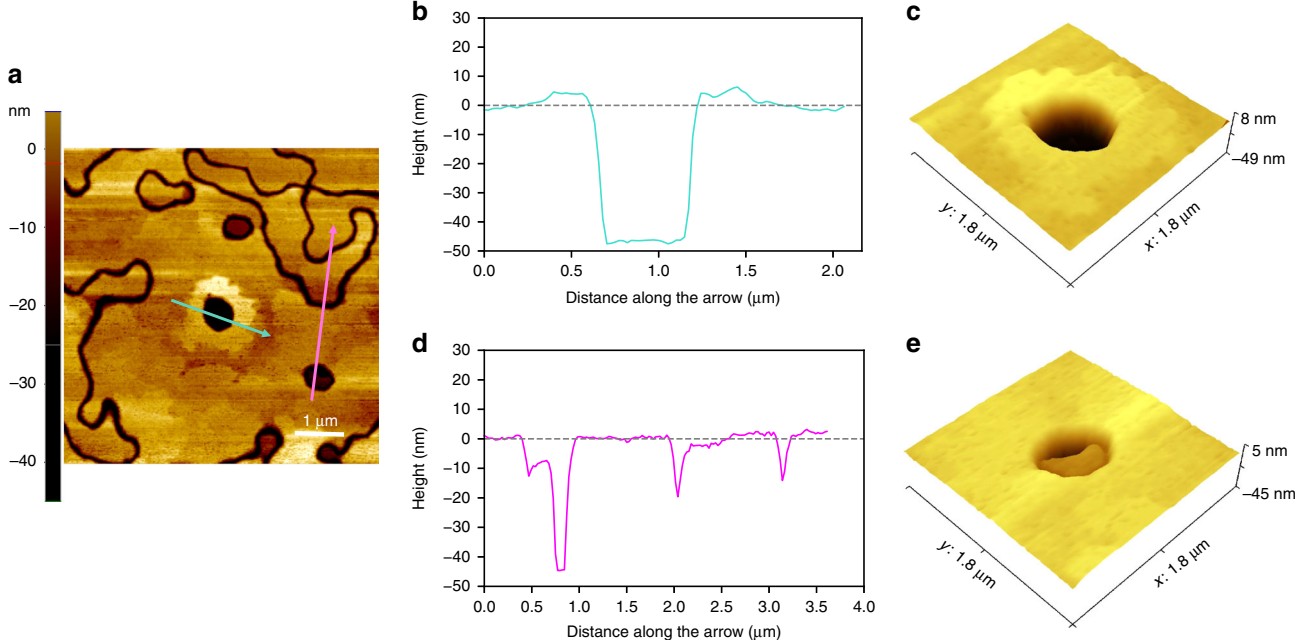

**Fig. 1** Experimental observations of an annealed Ni film on sapphire. **a** Atomic force microscopy (AFM) topography image of a Ni film on sapphire after heat treatment, showing sunken grains with (turquoise arrow) and without (magenta arrow) an elevated ridge around the edges. **b** A linear topographic profile across the hole indicated using a turquoise arrow in **a** showing a Ni ridge at the perimeter of the sunken grain. **c** 3-D image of the same hole showing an elevated ridge surrounding the sunken grain. **d** A linear topographic profile across the hole indicated using the magenta arrow in **a** showing no measurable ridge formation. **e** 3-D image around the same hole showing a depression left by grain sinking

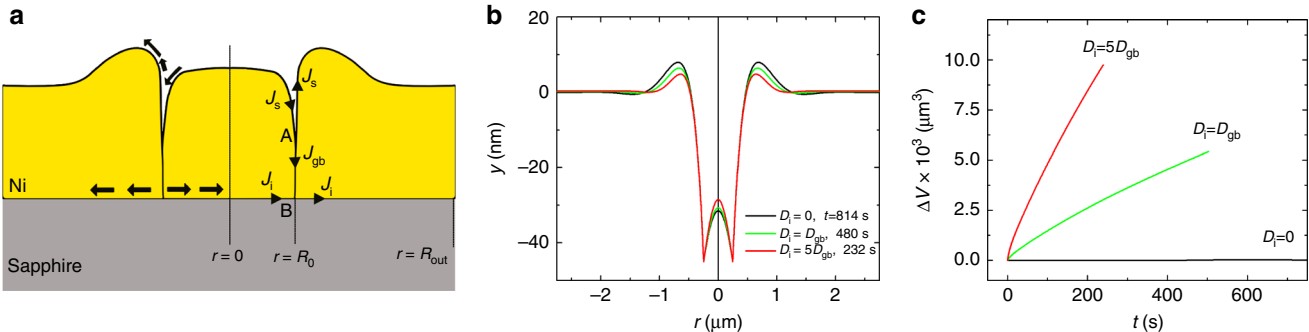

**Fig. 2** Development and results of the morphology evolution model. **a** The Ni/sapphire model showing Ni flux along the Ni surface ($J_s$), along the GB (A is the GB root position) between the sinking middle grain and the outer grain ($J_{gb}$), and along the Ni/sapphire interface ($J_i$) (B is the location of the intersection of the GB and the interface). **b** Surface topography profiles $y(r)$ for three different interface diffusivities shown in black, green and red at the time $t$ when the GB root hits the metal/ceramic interface at 700 °C. **c** Calculated "missing" Ni volume $\Delta V$ vs. time

This interface diffusivity is surprisingly large given that this metal/ceramic interface is nearly coherent and the strong bonding between this metal and ceramic. We now investigate the cause of this large interface diffusivity and discuss whether it is a generic finding for all metal/ceramic interfaces. (We note that oxygen transport may be neglected since the oxygen solubility in Ni is very low; the experiments were performed under the flow of forming gas and no oxygen-containing spinel phases were observed.)

**First-principles modeling of point defects**. To understand fast transport along the Ni/sapphire interface, we first focus on point defect formation energetics using density functional theory (DFT) (to account for the complex bonding at the interface). One of the earliest works that looked at the point defect formation energy near a metal/ceramic interface was carried out by Duffy et al.[28]

using a combination of first-principles and classical methods for Ag/MgO. More recently, there have been studies of oxygen vacancy formation energy[29,30] and oxygen Frenkel pair[31] formation energies at metal/oxide interfaces and oxide/oxide interfaces[32,33]. However, most such studies of metal/ceramic interfaces focused on the oxide side of the interface and on the electronic behavior of the contact rather than diffusion—the primary focus of the present work. There have also been no reports on point defect migration along the interface. Here, we carry out a systematic investigation of point defect energetics near the Ni/sapphire interface as well as diffusion along it.

We first turn to existing experimental observations to aid the construction of the interface in our computational model. Transmission electron microscopy examinations of the Ni/sapphire interface established two distinct orientation relationships. These are M1: Ni(111)[1$\bar{1}$0]||Al$_2$O$_3$(0001)[11$\bar{2}$0] and M2: Ni(111)[1$\bar{2}$1]||Al$_2$O$_3$(0001)[11$\bar{2}$0][34,35] (which is rotated by 30° about

the surface normal from M1)[35]. Perfect interface coherency demands that Ni is biaxially strained −2.8% for M1 and +12.3% for M2. We focus on the less strained M1 interface which is the only observed variant in our experiments (the M2 interface will have a much higher interface energy associated with a high density of misfit dislocations). The O-terminated $Al_2O_3$/Ni interface is shown in Fig. 3a; the $Al_2O_3$ may also be terminated by one or two Al atom planes. Examination of Fig. 3b shows that the 2Al-terminated and O-terminated interfaces are stable over a wide range of oxygen chemical potentials (Supplementary Discussions 1 and 2); at high (low) $p_{O_2}$ the O (2Al)-terminated Ni/$Al_2O_3$ interfaces will be thermodynamically stable (the maximum and minimum $p_{O_2}^{max}$ and $p_{O_2}^{min}$ are set by Ni oxidation and $Al_2O_3$ reduction (Supplementary Table 2). For each sapphire termination, the Ni (111) termination may be A, B or C, corresponding to the classical description of the (111) plane stacking of FCC materials (i.e., …ABCABC…)—these 3 terminations also represent Ni crystal shifts parallel to the interface (Supplementary Fig. 1). Our DFT calculations show that the termination C-Ni/sapphire is the most stable for both 2Al and O sapphire terminations. For the O-terminated sapphire interface, the C-Ni termination corresponds to placing a Ni atom at the same position that would be occupied by an Al atom in perfect sapphire[36].

Since diffusion in Ni is vacancy-controlled[37], we determine vacancy formation energies in Ni and $\alpha$-$Al_2O_3$ as a function of distance $d$ (Supplementary Fig. 2) from the Ni/sapphire interface and $p_{O_2}$; e.g., the formation energy of the neutral O-vacancy (i.e., formed by removing an O atom and all of its electrons) in $\alpha$-$Al_2O_3$ is denoted by $E_f^{V_O^\times}\left(d, p_{O_2}\right)$ in Kröger–Vink notation[38]. At

the metal/ceramic interface, we expect the net charge on point defects to be near zero since the Fermi level of the system will be pinned to that of the metal[39]. Hence, unlike in bulk ceramics (Supplementary Discussion 1, Supplementary Fig. 3), neutral point defects may be formed near both sides of the interface.

We first calculate neutral vacancy formation energies in bulk Ni and $\alpha$-$Al_2O_3$ ($d = \infty$); $E_f^{V_{Ni}^\times}(\infty, \cdot)$ ($p_{O_2} = $ "$\cdot$" indicates $p_{O_2}$-independence), $E_f^{V_{Al}^\times}\left(\infty, p_{O_2}\right)$ and $E_f^{V_O^\times}\left(\infty, p_{O_2}\right)$. $E_f^{V_{Ni}^\times}(\infty, \cdot) =$ 1.51 eV (unstrained, close to the experimentally found value of 1.6 eV[40]) or 1.25 eV (strained by −2.8% for M1 epitaxial relationship). Similarly, since Schottky defects are more prevalent than Frenkel defects in $\alpha$-$Al_2O_3$[41,42], we focus on Al and O vacancies on the sapphire side of the interface. For bulk sapphire, the neutral vacancy formation energies depend on oxygen partial pressure; on oxygen sites $E_f^{V_O^\times}\left(\infty, p_{O_2}^{min}\right) = 1.92$ eV and $E_f^{V_O^\times}\left(\infty, p_{O_2}^{max}\right) = 6.18$ eV. The neutral Al vacancy formation energy is $E_f^{V_{Al}^\times}\left(\infty, p_{O_2}^{min}\right) = 13.85$ eV and $E_f^{V_{Al}^\times}\left(\infty, p_{O_2}^{max}\right) = 7.46$ eV (Supplementary Fig. 3). These vacancy formation energies suggest that near the interface the equilibrium vacancy concentration in Ni is much higher than either Al or O vacancies in sapphire.

Figure 4 shows the neutral Ni, Al, and O vacancy formation energies as a function of distance $d$ from the O-terminated sapphire and 2Al-terminated sapphire/C-plane Ni interfaces. For the O-terminated sapphire interface, the Ni vacancy formation energy $E_f^{V_{Ni}^\times}(d, \cdot)$ drops from its bulk value 1.25 eV far from the interface to 0.85 eV (i.e., 68% of the bulk value for pure Ni) two (111) Ni atomic planes from the interface. On the other hand,

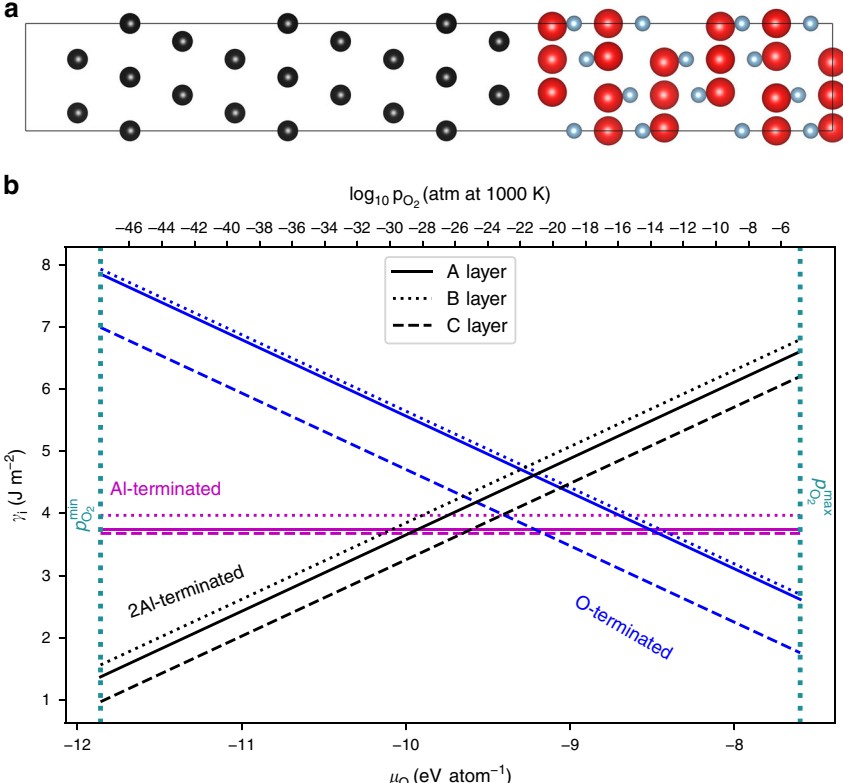

**Fig. 3** DFT prediction of the M1 Ni(111)$[1\bar{1}0]$||$Al_2O_3$(0001)$[1\bar{1}\bar{2}0]$ interface energy. **a** Unrelaxed, atomistic model of the (C-terminated) Ni/(O-terminated) sapphire interface. Ni, Al, and O are shown in black, blue and red. **b** Interface energy of the Ni/sapphire interface considering the three Ni translations (A, B, C) and the three sapphire terminations (2Al-terminated, Al-terminated, O-terminated shown in black, magenta, and blue, respectively) within the limits of $p_{O_2}^{max}$ and $p_{O_2}^{min}$ shown using dotted vertical teal lines

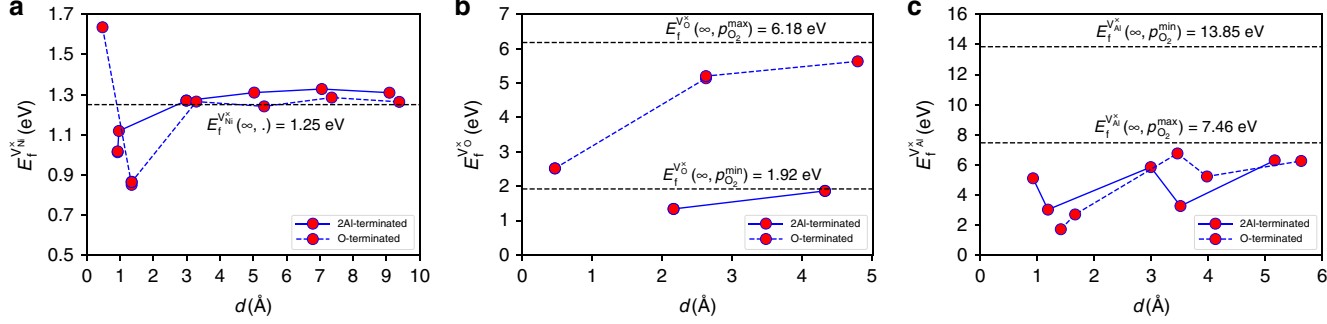

**Fig. 4** Variation of vacancy formation energies $\left(E_f^{V_{Ni}^\times}, E_f^{V_O^\times}, E_f^{V_{Al}^\times}\right)$ versus distance from the interface $d$. **a** Ni vacancy formation energy for both O-terminated and 2Al-terminated sapphire interfaces. The horizontal dashed line represents the vacancy formation in bulk strained Ni (corresponding to M1 epitaxy). **b** Neutral O vacancy formation energy for both the terminations. For the O-terminated case, the bulk O vacancy formation energy (dashed line) is for $p_{O_2}^{max}$ and at $p_{O_2}^{min}$ for the 2Al-termination. **c** Neutral Al vacancy formation energy for both the terminations. The bulk Al vacancy formation energies for the O-terminated and 2Al-terminated cases are for $p_{O_2}^{max}$ and $p_{O_2}^{min}$

**Table 1 Predictions of the descriptor compared with DFT results**

| | DFT | | | Descriptor |
|---|---|---|---|---|
| | $E_f^{V_m^\times}(0)$ **(eV)** | $E_f^{V_m^\times}(\infty)$ **(eV)** | $\dfrac{E_f^{V_m^\times}(0)}{E_f^{V_m^\times}(\infty)}$ | $\dfrac{W_{ad}}{\gamma_m}$ |
| Ni | 0.85 | 1.25 | 0.68 | 0.62 |
| Cu | 0.25 | 0.56 | 0.45 | 0.37 |
| Ti | 2.00 | 2.19 | 0.91 | 0.94 |

Vacancy formation energies of Ni, Cu, and Ti at their interfaces with sapphire and in their bulk strained states, their ratio and the prediction as per Eq. (1) from refs [47,48,49]

immediately adjacent to the sapphire in Fig. 4a, the Ni vacancy formation energy is very large, 1.64 eV, likely because of the strong metal–oxygen bond. While removing a Ni atom from this site is not as energetically costly as removing an Al atom from bulk sapphire (the Al vacancy formation energy is 7–13 eV depending on $p_{O_2}$), it is higher than it would be for removing a Ni atom from a Ni crystal. The same trend also applies to the 2Al-terminated sapphire interface. Hence, the Ni vacancy formation energy is much smaller near the interface than in the Ni interior for all Ni/$\alpha$-Al$_2$O$_3$ interfaces, implying the thermal concentration of Ni vacancies near the interface is much higher than elsewhere in Ni.

Figure 4b, c shows that the O and Al vacancy formation energies decrease from their bulk values as we near the interface. For example, in the O-terminated sapphire case, the O vacancy formation energy decreases from 5.63 to 2.52 eV as it approaches the interface (at $p_{O_2}^{max}$), while for the 2Al-terminated case, it drops from 1.86 eV in the bulk to 1.34 eV near the interface at $p_{O_2}^{min}$. Previous studies of O vacancy formation energy near Pt/TiO2[29] and (Pt,Ag,Al)/HfO2[30] interfaces show a similar trend of decreasing O vacancy formation energy on approaching the interface. Similarly, looking at the Al vacancy in the O-terminated case, $E_f^{V_{Al}^\times}\left(d, p_{O_2}^{max}\right)$ drops from 6.25 to 1.71 eV near the interface. We however note that while the Al vacancy formation energy $E_f^{V_{Al}^\times}$ in 2Al-terminated sapphire case is relatively small near the interface, it does not appear to approach its bulk value $E_f^{V_{Al}^\times}\left(\infty, p_{O_2}^{min}\right)$ far into the sapphire. Examination of the atomic structure of this vacancy shows that it is different from that observed in bulk sapphire[43], even at the maximum separation from the interface in our computational cell ($\sim$6 Å).

While the lowest formation energy point defect near the interface is the Ni vacancy $E_f^{V_{Ni}^\times} = 0.85$ eV, the third lowest is the Al vacancy in O-terminated sapphire $E_f^{V_{Al}^\times} = 1.71$ eV (the second lowest energy is an oxygen vacancy). In this case, however, the Al vacancy is replaced by a Ni interstitial and a vacancy on the Ni side of the interface (Supplementary Fig. 4). Hence, consideration of defect complexes at the interface involving the low-formation energy defects $\left(V_{Ni}^\times, V_{Al}^\times\right)$ reveals that the Ni vacancy concentration at the metal/ceramic interface is expected to be even higher than that suggested by the single point defect formation energies alone.

**Interface transport**. Vacancy defect-mediated diffusion is commonly characterized as $D = D_0 e^{-E_f^V/k_B T} e^{-E_m^V/k_B T}$, where the pre-exponential factor $D_0$ accounts for crystal structure, the effective atomic vibration frequency, interatomic separation, correlation and entropy effects, $k_B T$ is the thermal energy, $E_f^V$ and $E_m^V$ are the vacancy formation and migration energies respectively[44]. The arrhenius terms describe the equilibrium vacancy concentration and vacancy migration, respectively. While in ceramics, the point defect density may be modified by doping, in metals it is usually dictated by equilibrium thermodynamics (vacancies are easily produced/annihilated by dislocation climb).

Using nudged elastic band calculations[45], we determine the barrier for Ni vacancy migration (Supplementary Fig. 5, Supplementary Table 3) parallel to the interface (second layer) to be 0.49 eV, which is $\sim$1/2 its bulk value. Since the Ni vacancy formation energy at this location is $\sim$0.7 that in bulk Ni, our results are consistent with earlier discovered trends[46] that showed that in elemental FCC and HCP metals both the vacancy formation and migration energies scale in the same manner (linearly) with cohesive energy (i.e., the drop in the vacancy energies is related to reduced cohesion at the interface compared with bulk Ni). Combining the vacancy formation and migration results reported here, these results suggest that at, for example, half the Ni melting point, the interface diffusivity should be $\geq 10^4$ times faster than in bulk Ni. GB diffusivities in metals are typically $(10^4–10^6)$ faster than lattice diffusion[14], which implies that metal/ceramic interface diffusivity and GB diffusivities are comparable at the same homologous temperature. Therefore, these results demonstrate that Ni transport along the Ni/sapphire interface is extraordinarily fast (relative to bulk diffusion). This conclusion is valid for both the coherent interface case analyzed here in detail and the case where the interface is semicoherent (the misfit dislocations in semicoherent interfaces are short-circuit diffusion paths).

To explore the generalization of this result, we compare the formation energy of a vacancy at the metal/ceramic interface to that within the bulk metal in terms of the local bonding at these locations, as captured by the metal/ceramic work of adhesion $W_{ad}$ and the metal surface energy $\gamma_m$. We estimate the ratio of the metal vacancy formation energy at the metal/ceramic interface to that in the bulk metal in terms of a simple, heuristic bond breaking model (Supplementary Method 2) as

$$\frac{E_f^{V_m^\times}(0)}{E_f^{V_m^\times}(\infty)} = \frac{W_{ad}}{\gamma_m}. \qquad (1)$$

To test the applicability of this simple prediction, we compare it with the DFT results for FCC-Ni/$\alpha$-Al$_2$O$_3$ (as described above), FCC-Cu/$\alpha$-Al$_2$O$_3$ and HCP-Ti/$\alpha$-Al$_2$O$_3$. As shown in Table 1 and Fig. 5, the empirical descriptor ($W_{ad}/\gamma_m$) accurately predicts the ratio of $E_f^{V_m^\times}(0)/E_f^{V_m^\times}(\infty)$ to within 3% for Ti, 18% for Cu, and 10% for Ni. This is remarkable agreement given the simplicity of Eq. (1).

## Discussion

We use the obtained descriptor to predict the ratio of the vacancy formation energy at the metal/ceramic interface to its value in the bulk metal for a wide range of metal/ceramic systems based upon Eq. (1) and using data readily available from the literature[47–52]. The resultant predictions are summarized in Fig. 5. Given the correlation between activation energy for vacancy migration and vacancy formation, we predict that metal diffusion along the interface will be faster than bulk diffusion for systems in which $E_f^{V_m^\times}(0)/E_f^{V_m^\times}(\infty)<1$. Most of the metal/ceramic systems shown in Fig. 5 fall into this category, including Ni/Al$_2$O$_3$ (and Cu/Al$_2$O$_3$). For systems with $E_f^{V_m^\times}(0)/E_f^{V_m^\times}(\infty) \sim 1$, diffusion in the bulk and at the interface will be comparable (Ti/Al$_2$O$_3$) and for others $\left( E_f^{V_m^\times}(0)/E_f^{V_m^\times}(\infty)>1 \right)$ interface diffusion is slower than in the bulk. The case of Ti/Al$_2$O$_3$, for which we have DFT data, is near the cusp—the interface diffusivity should be comparable with bulk diffusion in Ti. Amongst the cases shown in Fig. 5 are many metal/ceramic pairs that are commonly used across a wide range of technologies. We note that the {Cr, Mn, Au, Cu, Sn}/Al$_2$O$_3$ and that {Au, Cu, Sn, Ag, Fe, Co, Ga}/SiO$_2$ interfaces all show $E_f^{V_m^\times}(0)/E_f^{V_m^\times}(\infty)<0.5$, suggesting that all of these systems will exhibit extremely fast metal transport along the metal/ceramic interfaces. However, given that extremely fast metal atom diffusion along metal/ceramic interfaces is the rule rather than the exception, fast metal/ceramic interface diffusion should not be considered anomalous after all.

We thus show both experimentally and computationally that diffusion at the Ni/sapphire interface is surprisingly fast. Our first-principles vacancy formation and migration energy calculations demonstrate that this is a result of relatively low cohesion at this interface compared with bulk Ni. This observation suggests a simple descriptor for diffusion at the interface compared with the bulk based upon readily available experimental and/or first-principles results. Based on this descriptor, we conclude that for most metal/ceramic systems (we examined close-packed metals and sapphire, silica, zirconia), interface diffusion is fast compared with the bulk and comparable with metal grain boundary diffusivities in many cases; yet there are exceptions (as determined based on interface cohesion). Systems where the metal only weakly wets (or does not wet) the ceramic, the interface diffusivity will be high; inversely, where the tendency for wetting is strong, the interface diffusivity will be low (all relative to the bulk metal). This suggests that alloying to modify wettability also affects

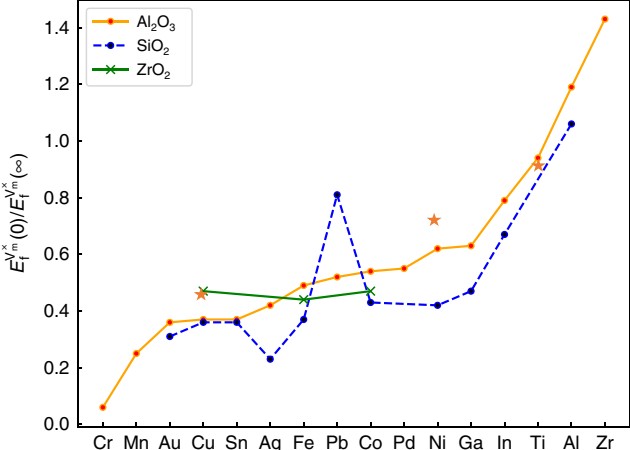

**Fig. 5** The predicted ratio of the vacancy formation energies at the metal/ceramic interface to that in the bulk metal for several metal/ceramic systems according to Eq. (1). The values for $\gamma_m$ and $W_{ad}$ are from[47–52]. The metals are indicated along the horizontal axis and different ceramics by the three curves. The Ni/Al$_2$O$_3$, Cu/Al$_2$O$_3$, and Ti/Al$_2$O$_3$ data (orange stars) are from direct DFT calculations

interface diffusion kinetics. This simple result provides easily applicable guidance for material design in a wide range of applications; especially in the energy and microelectronics industries.

## Methods

**Sample preparation.** 45 nm thick Ni films (5N purity) were deposited by electron beam deposition on (0001) c-plane-oriented $\alpha$-Al$_2$O$_3$ (sapphire) single crystal substrates. Prior to deposition, the substrates were prepared in a clean room with soap, deionized water, acetone, methanol, isopropanol, followed by dipping in Piranha solution (H$_2$SO$_4$ and H$_2$O$_2$, 2:1 volume ratio) and deionized water. The films were deposited at a rate of 1 Å s$^{-1}$ at room temperature and a vacuum of 7 × 10$^{-7}$ torr. The deposition resulted in a maze-bicrystal microstructure. The samples were then annealed for 10 min at 700 °C in forming gas flow (Ar-10 vol% H$_2$, 5N purity) in a rapid thermal annealing (RTA) furnace.

**Characterization.** The morphology of the sample was characterized by atomic force microscopy (AFM; Park systems XE-70) in tapping mode using a Si tip with a typical radius of curvature of 10 nm. The scan rate was 0.2–0.7 Hz.

**Short-circuit diffusion model.** The evolution of the surface topography was modeled employing the axisymmetric variational scheme described in detail in ref. [53]. The method is based on surface discretization and a variational calculation of surface chemical potentials first proposed by Dornel et al.[54] for a one-dimensional surface. A detailed description of the formulation of the model specifically for these applications is presented in the Supplementary Method 1.

**First-principles calculations.** All DFT calculations were performed using the Vienna Ab initio Simulations Package (VASP)[55–58]. The core electrons were described using the Projector Augmented Wave (PAW) scheme[59,60] and exchange-correlation was described using the recently developed meta-GGA functional SCAN[61], which was shown to perform better than PBE-GGA[62] for diversely bonded systems. The energy cut-off for the plane wave expansion was 520 eV for all calculations except those performed on pure metallic systems of Ni, Cu, and Ti where the cut-off was 350 eV.

For the single crystal metal supercell calculations, a Monkhorst-Pack[63] K-mesh was used which was 9 × 9 × 3 for Ni and Cu, and 12 × 12 × 3 for Ti. The supercell size for Ni and Cu was 24 atoms while for Ti, a supercell of 18 atoms was used. The vacancy migration barrier calculations in pure Ni were performed using the climbing image-nudged elastic band approach[45] in a supercell containing 47 atoms (containing a Ni vacancy), previously relaxed using a 3 × 9 × 3 K-mesh (Monkhorst-Pack). The bulk vacancy formation and barriers calculations were performed in supercells strained to match that of the metal in the metal/ceramic interface supercells. The 30-atom sapphire supercell (6 $\alpha$-Al$_2$O$_3$ formula units) was volume relaxed (Supplementary Fig. 6, Supplementary Table 4) using a $\Gamma$ centered K-mesh of 9 × 9 × 3.

The Ni/sapphire supercell consisted of 6 $\alpha$-Al$_2$O$_3$ formula units and 28, 36, and 44 Ni atoms (various Ni terminations). The O-terminated and 2Al-terminated

sapphire were created by sequentially removing an Al-plane and an O-plane respectively. 36 Cu atoms and 21 Ti atoms were used for the Cu/sapphire and the Ti/sapphire interfaces respectively, with the same number of atoms on the sapphire side as in the Ni/sapphire case. For the Ti/sapphire interface, the bicrystallography determined from experiments[64] was used. For all these interface systems, a Γ centered K-mesh of $9 \times 9 \times 1$ was used.

For the Ni/sapphire migration barrier calculations, the supercell was two times larger in the $b$ direction (Supplementary Fig. 5) and contained 127 atoms (relaxed using a Γ centered K-mesh of $9 \times 4 \times 1$). We allowed 5 atomic layers around the interface to relax while those more distant from the interface were frozen at their perfect crystal positions. For the point defect formation energy calculations in (otherwise) perfect $\alpha$-Al$_2$O$_3$, a supercell of 120 atoms was used with a a Γ centered K-mesh of $4 \times 4 \times 2$.

## Data availability

Any associated data and code are available upon reasonable request from the authors, refer to Author Contributions for relevant data.

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

## Acknowledgements

The authors gratefully acknowledge the support of the (A.K. and D.J.S.) US National Science Foundation grant 1609267, (H.B. and E.R.) US-Israel Binational Science Foundation grant 2015680 and Israel Science Foundation grant 1628/15. Work partly performed under the auspices of US DOE by LLNL under Contract DE-AC52-07NA27344. The authors acknowledge fruitful discussions with Jian Han, Jianwei Sun, Anuj Goyal, Dor Amram, Bilge Yildiz and Mostafa Youssef, and Christoff Freysoldt.

## Author contributions

A.K., E.R., and D.J.S. designed the research; A.K. and H.B. performed the computational and experimental research respectively; L.K.and E.R. developed the diffusion model; L.K. performed the numerical calculations, M.W.F. and V.L. contributed to discussions and analysis; the diffusion descriptor was proposed by A.K, E.R., and D.J.S; and A.K., H.B., E.R., D.J.S. wrote the paper.

## Additional information

**Competing interests:** The authors declare no competing interests.

