## [Peer Review File · Nature Communications]

Reviewers' comments:

Reviewer #1 (Remarks to the Author):

In this manuscript, the authors study the Ni/ α -Al₂O₃ interface via AFM measurements and theoretical modelling. They find features in thin films of Ni over a Al₂O₃ substrate that can be explained by fast diffusion of Ni atoms along the interface. The experimental setup looks well done to me, the diffusion model is well thought and the ab initio calculations that support it are done with a good choice of functional. The conclusions are original and interesting for the wide condensed matter physics community, as they provide a generalization to the conditions for fast diffusion behaviour of these types of interface.

My main suggestion for improvement comes from looking at Figure 5 in the Supplementary Information. I find it difficult to see the path of the diffusing atom along the interface in this picture.

There are also several typos in the manuscript. The most blatant case is in Page 10 of the Supplementary Information ("balance band").

After corrections are done, I recommend publishing this work.

Reviewer #2 (Remarks to the Author):

Report on the manuscript

Anomalous Diffusion along Metal/Ceramic Interfaces
by Kumar et al.

This paper presents a careful in-depth analysis of the diffusion processes taking place at metal/ceramic interfaces during annealing at elevated temperatures below the melting point of the metal. As a case study, the hole formation in a polycrystalline Ni thin film deposited on a (0001) oriented single crystalline alumina substrate was studied. The experimental observations are explained with the help of a continuum diffusion model and ab-initio calculations. The latter revealed that diffusion along a coherent Ni/alumina interface is faster than in bulk Ni and similar to the diffusion along large angle grain boundaries, an unexpected and very important result. Hole formation is critical for the life time of metallic conductor lines as it leads to catastrophic failure of the devices. In contrast, hole formation can be also used in a positive way for nano-structuring of metal surfaces. In any case, a better understanding of the involved diffusion processes can have significant impact on either minimizing hole formation or for targeted patterning of metals. The paper is very well written and clearly structured. I have only some minor suggestions which the authors should consider in a revised manuscript:

- The authors focus in their work on the contribution of the interface diffusion at the metal/ceramic interface although they mention that within the continuum model "simultaneous surface, GB, and interface metal diffusion" (page 2) is taken into account. A short discussion on the possible contribution of surface diffusion should be added as no passivating layer is deposited on top of the Ni film which could minimize surface diffusion.
- The authors state on page 5 that "This conclusion is valid for both the coherent interface case analyzed here in detail and the case where the interface is semicoherent (which should be faster)." In the case of the semicoherent interface strong strain fields will develop around the misfit dislocation – as such it is not obvious why the diffusion should be faster for this case. Please explain.
- What is the solubility of oxygen in Ni? Can O also diffuse from the environment and/or the alumina through the film and has this an impact on the hole formation?

- To further support the statement that the individual grains sink and disappear completely, a cross-section of this region would be beneficial. How can they prove that not a few atomic layer thin Ni film is still present on the alumina substrate?
- As stated, the misfit for the orientation relationship M2 with 12.3% is quite high and difficult to account for in the ab-initio calculations. However, in the experiments, the two interface configurations can be distinguished in cross-sectional TEM micrographs. Did they observe a change in thickness around the holes only for those grains possessing the orientation relationship M1 (coherent interface)?
- In Figure 4 it is not clear what is the exact position of "O"?
- Two values are given for the Ni vacancy formation energy in the bulk (1.28 eV and 1.25 eV (strained)), which one is the correct one?
- On page 6 the authors state that "given that extremely fast metal atom diffusion along metal/ceramic interfaces is the rule rather than the exception, fast metal/ceramic interface diffusion should not be considered anomalous after all." How does this statement fit to their title: "Anomalous Diffusion along Metal/Ceramic Interfaces". Maybe they want to change this?

Reviewer #3 (Remarks to the Author):

According to abstract, this manuscript aims to present generalized model for heterogeneous interface between ceramic materials and metallic materials, which is critically important for not only for the community of grain boundaries/interfaces but also applications of which "performance"s are dominated by the diffusion along the interfaces.

However, I cannot suggest Editor to accept this manuscript for publication in the currently prepared form, although I believe that a study behind the manuscript is of interest for possible readers of this journal.

Major reasons behind my suggestion are as follows (written only in concise manner):

1. Introducing the importance of this heterogeneous interfaces and diffusion along them in the introductory paragraphs are not general enough for general readers or carefully enough for experts, either. Especially, what is "direct" evidence is not clearly stated in the introduction, and how the diffusion along the heterogeneous interfaces affect "performance"s of applications exemplified by the authors is not clear at all: There is a leap in logics, though I myself can "guess" how important it will be. The authors should restructure this section such that readers, both experts and non-experts can focus on the focal point of this study behind the manuscript.
2. Before this study, the studies that have reported that vacancy formations at or in the vicinity of the heterogeneous interface exist but they are not cited at all, by stating almost like that there is no preceding studies that reports ab initio calculations to examine factors affecting diffusion along the heterogeneous grain boundaries.
3. What presented based on transmission electron microscopy observation appears to be "indirect" evidence of the fast diffusion, although I do not underestimate its importance, validity or usefulness as compared with in-situ STEM observations. This section is against what the authors claim in the introductory section.
4. Quantifications by ab initio calculations are nothing better than ordinary calculations frequently appear in other journals. I am not convinced by this section of the manuscript that the authors have gained deeper insights for diffusion along the heterogeneous interfaces. In addition, I cannot find the justification well discussed to generalize the findings for a specific heterogeneous interface toward generalized heterogeneous interfaces between ceramic materials and metallic materials, to

draw conclusions given in the abstract.

For these reasons, I cannot recommend publication of the manuscript in the present form unless it is designed again and re-written, although, again, I believe the study behind the manuscript has scientific significance and novelties if and only if it is well designed and written.

End of comments.

Responses to Reviewers' comments

Reviewer 1:

- 1. My main suggestion for improvement comes from looking at Figure 5 in the Supplementary Information. I find it difficult to see the path of the diffusing atom along the interface in this picture.**

Response: We agree that Fig. 5 in the Supplementary Information was somewhat confusing. We have replaced that figure with a new one which shows the path of the Ni vacancy much more clearly.

- 2. There are also several typos in the manuscript. The most blatant case is in Page 10 of the Supplementary Information ("balance band")**

Response: We thank the reviewer for catching this (embarrassing) typo. We have now gone through the manuscript and the Supplementary Information and re-spell checked both documents.

Reviewer 2:

- 1. The authors focus in their work on the contribution of the interface diffusion at the metal/ceramic interface although they mention that within the continuum model "simultaneous surface, GB, and interface metal diffusion" (page 2) is taken into account. A short discussion on the possible contribution of surface diffusion should be added as no passivating layer is deposited on top of the Ni film which could minimize surface diffusion.**

Response: We fully agree with the reviewer that diffusion on the surface of the Ni film should be active in our experiment (otherwise, no film morphology changes would be possible at all). Our point is that surface diffusion alone cannot explain the mass imbalance that occurs near the holes in the film. We demonstrate that the "missing" Ni (i.e. Ni that is not leading to surface-diffusion-controlled surface ridge formation) is transported by a combination of grain boundary and interface diffusion. We note that Ni atoms diffuse along the surface of the film, arrive at the GB, and then diffuses down the GB to the metal/ceramic interface. While classical models of surface diffusion-controlled grain boundary grooving and ridge formation still operate, they imply surface diffusion away from the GB while in the present grain sinking case, the surface diffusion is in the opposite direction (towards the GB). We add a short version of this discussion in the revised manuscript (see the 2nd and 3rd paragraphs of the "Results" section).

- 2. The authors state on page 5 that "This conclusion is valid for both the coherent interface case analyzed here in detail and the case where the interface is semicoherent (which should be faster)." In the case of the semicoherent interface strong strain fields will develop around the misfit dislocation – as such it is not obvious why the diffusion should be faster for this case. Please explain.**

Response: While we agree with the reviewer that strong elastic strain fields develop in the vicinity of misfit dislocations, the (misfit) dislocation cores represent short circuit diffusion paths that generally

have lower activation energies for diffusion than the surrounding material. (We note that dislocations are always sources of stress, yet dislocations are widely known as fast diffusion paths.) Since Ni is under compression in the coherent system, the misfit dislocations will be of net edge character with the Burgers vector such that the low-density region around the dislocation core will be on the Ni side of the interface. This implies that misfit dislocations should increase the net diffusivity of the semicoherent interface over that of its coherent counterpart.

3. *What is the solubility of oxygen in Ni? Can O also diffuse from the environment and/or the alumina through the film and has this an impact on the hole formation?*

Response: The solubility of oxygen in solid Ni was determined by Park and Altstetter (1987)¹ in the 800-1000° C temperature range. Extrapolation of their data to 700° C (our experimental condition) yields an oxygen solubility of 90 ppm. Although this amount of oxygen can affect diffusion rates, it is much too small to cause any mass imbalance in the vicinity of the hole. Also note that our experiments were performed in flowing high-purity forming gas, which should minimize the exposure of the sample to oxygen during annealing. Finally, the absence of oxygen-containing spinel phases at the Ni/sapphire interface as observed in the transmission electron microscope (TEM) further supports the idea that oxygen does not play a role in hole formation. We summarize these points at the end of the “Experimental evidence of fast interface diffusion” section.

4. *To further support the statement that the individual grains sink and disappear completely, a cross-section of this region would be beneficial. How can they prove that not a few atomic layer thin Ni film is still present on the alumina substrate?*

Response: First, we note that the clear experimental observation that some Ni grains sink and (completely or nearly completely) disappear is sufficient to demonstrate that interface diffusion is very fast - this does not require that the sinking grain completely disappears. Yet, we cannot claim with 100% confidence that a thin (one-two monolayer) Ni layer is not present on the sapphire substrate in place of a sunken grain.

The absence of such layer on the scanning TEM (STEM) image of the hole cross-section may be misleading because preparing such site-specific samples in a focused ion beam (FIB) instrument involves the deposition of protective C and Pt layers. The exposure of the sample to the flux of these atoms could, in principle, destroy such a very thin surface layer of Ni. Nevertheless, a combination of atomic force microscopy (AFM) data demonstrating that the depth of the hole corresponds to the thickness of the deposited film, and the STEM images (shown here) lends credibility to the hypothesis that the sinking Ni grain disappears completely, without leaving traces of Ni on the substrate. We present below a STEM cross-sectional image of a bigger hole with well-developed rims in a similar sample (40 nm-thick Ni film on sapphire annealed for 6 min at the temperature of 700 °C). Energy dispersive spectroscopy (EDS) elemental maps do not show any Ni at the bottom of the hole (the magnified region of the hole adjacent to the rim is shown in the bottom part of the figure).

5. *As stated, the misfit for the orientation relationship M2 with 12.3% is quite high and difficult to account for in the ab-initio calculations. However, in the experiments, the two interface configurations can be distinguished in cross-sectional TEM micrographs. Did they observe a change in thickness around the holes only for those grains possessing the orientation relationship M1 (coherent interface)?*

Response: The experimental observations point to the existence of **two different variants of the M1 interface** that are rotated by 60° to their surface normal. We never observe the M2 interface in our experiments. We have added a sentence to clarify this in the second paragraph of the “First-Principles modeling of point defects” section.

6. In Figure 4 it is not clear what is the exact position of “O”?

Response: We agree that there is some ambiguity in identifying the exact position d for any of the atoms in the system. To remove this ambiguity, we have added Supplementary Fig. 2 in the Supplementary Information document that shows both the relaxed 2Al- and O-terminated interfaces and explicitly gives the definition of $d=0$.

7. Two values are given for the Ni vacancy formation energy in the bulk (1.28 eV and 1.25 eV (strained), which one is the correct one?

Response: The correct value is 1.25 eV. We thank the reviewer for pointing out an erroneous value in the text. This has been corrected in the manuscript (third to last paragraph of the “First-principles modeling of point defects” section).

8. On page 6 the authors state that “given that extremely fast metal atom diffusion along metal/ceramic interfaces is the rule rather than the exception, fast metal/ceramic interface diffusion should not be considered anomalous after all.” How does this statement fit to their title: “Anomalous Diffusion along Metal/Ceramic Interfaces”. Maybe they want to change this?

Response: Our point in choosing this title is that the diffusivity is anomalous in terms of what is widely expected. We do realize that the conclusion suggests that this expectation often fails; that fast diffusion is not unusual (this is the basis of the reviewer’s comment). Hence, “anomalous” depends on with respect to what. We do appreciate the Reviewer leaving this to our decision.

(We do note that our analysis does indeed show that metal/ceramic interfaces can, indeed, have a wide range of diffusivities and we have identified systems where it is both very fast and very slow.)

Reviewer 3:

1. Introducing the importance of this heterogeneous interfaces and diffusion along them in the introductory paragraphs are not general enough for general readers or carefully enough for experts, either. Especially, what is “direct” evidence is not clearly stated in the introduction, and how the diffusion along the heterogeneous interfaces affect “performance”s of applications exemplified by the authors is not clear at all: There is a leap in logics, though I myself can “guess” how important it will be. The authors should restructure this section such that readers, both experts and non-experts can focus on the focal point of this study behind the manuscript.

Response: We have extensively re-written the “Introduction” section, and carefully selected a few experimental examples that clearly demonstrate the importance of metal diffusion along the metal ceramic interfaces. We have also removed other, more indirect examples. Perhaps, we did leave too much to the reader to fill in. See the first paragraph of the revised manuscript.

We also briefly explain (second paragraph of the revised manuscript) what constitutes a direct measurement of interface diffusion, the fact that such measurements do not exist for metal/ceramic interfaces and then go on to discuss a few indirect observations. We are now very careful to describe what is direct and what is indirect.

- 2. Before this study, the studies that have reported that vacancy formations at or in the vicinity of the heterogeneous interface exist but they are not cited at all, by stating almost like that there is no preceding studies that reports *ab initio* calculations to examine factors affecting diffusion along the heterogeneous grain boundaries.**

Response: It was not our intention to imply that nobody has previously done *ab initio* calculations of point defects near interfaces. Rather, we believe that the extant literature is only marginally relevant to the topic of this paper – metal transport along metal/ceramic interfaces. Nonetheless, we should have done a better job putting our calculations in a broader context. We have now added references to previous works²⁻⁷ that have looked at vacancy formation energies in the vicinity of heterogeneous interfaces including one of the earliest reported works of Duffy *et al.*² for the Ag/MgO interface. We also mention two studies on O vacancy formation energies near Pt/TiO₂³ and several metal/HfO₂⁴ interfaces, which show that the O vacancy formation energy decreases on approaching the interface. We also note in the manuscript that these studies have largely focused on the O vacancies or O Frenkel pair⁵ for heterogeneous interfaces including ceramic-ceramic interfaces^{6,7}, and not on the diffusion along these interfaces. These are added in the first and the penultimate paragraph under “First-principles modeling of point defects” section.

- 3. What presented based on transmission electron microscopy observation appears to be "indirect" evidence of the fast diffusion, although I do not underestimate its importance, validity or usefulness as compared with in-situ STEM observations. This section is against what the authors claim in the introductory section.**

Response: As explained in the revised version of “Introduction” section, direct measurements of diffusion along the metal-ceramic interface are both difficult and have not been made (to our knowledge) on well-characterized, defect-free bi-crystalline samples. The low density of defects on the metal side of the interface is especially important since otherwise the interface diffusion would be easily masked by dislocation pipe or subgrain boundary diffusion. We do not claim in our manuscript that we are making direct measurements of diffusion along the metal-ceramic interface – but rather provide, to paraphrase the reviewer, our AFM (not STEM or TEM as the reviewer states) results are “important, valid, and useful indirect evidence” of fast diffusion along metal/ceramic interfaces. Our measurements are indeed indirect but are close to the ideal since only one type of extraneous defect is present in our Ni film (special Σ 3 GB, where Σ is a reciprocal density of coincident sites in two misoriented lattices), and these defects are separated by distances much larger than the film thickness. This allowed us to build a simple diffusion model relating the rate of “sinking” of small circular grains with the interface diffusion. We would like to emphasize that similar methods of relating the changes of geometrical parameters of the system (e.g., precipitate size) with the diffusivity of defects was employed in determining the diffusion coefficient along the dislocation cores⁸.

- 4. Quantifications by *ab initio* calculations are nothing better than ordinary calculations frequently appear in other journals. I am not convinced by this section of the manuscript that the authors have gained deeper insights for diffusion along the heterogeneous interfaces. In addition, I cannot find the justification well discussed to generalize the findings for a specific heterogeneous interface toward generalized heterogeneous interfaces between ceramic materials and metallic materials, to draw conclusions given in the abstract.**

Response: We are not aware of any other work performing first-principles calculations of both point defect formation energy near metal-ceramic (especially on the relevant metal side) interface and the barriers for point defect migration along the metal-ceramic interface. Nonetheless, the specific details of the calculations are less important than the results that they yield. Metal vacancy formation and migration energies are very low near this interface – explaining the main experimental result. We believe another key advance in this paper, overlooked by the Reviewer, is that we were able to use these results to develop a simple model for many metal/ceramic systems and to validate the model based upon additional first-principles results (not used in developing the model). In our opinion, it is this simple-to-apply model that is the main outcome of this paper. Granted, to get the data that led to the model, we use standard methods, applied to a new problem (metal diffusion along metal/ceramic interfaces). In our view, this does not detract from the work. The new science is based on what we have been able to deduce from the results, not any particular raw number or application of a particular technique.

Differences between the present work and Amram *et al.*⁹

The earlier work of Amram *et al.*⁹ (Ref. [23] in the original manuscript and now Ref. [25] in the revised manuscript) should be viewed as a preliminary analysis of nearly the same system examined experimentally here (there were slight differences in the experiment). The focus of the earlier work was on understanding the observed (unusual) grain boundary (GB) grooves. Besides the difference in focus, the results from Ref. [23] were essentially one-dimensional – both in terms of the experimentally measured GB groove topography profiles (acquired as line profiles along the local normal to the GB trace), and the respective diffusion model. This one-dimensionality ignored the maze-like, curved in-plane shape of the GBs and possible contributions of surface diffusion in the direction along the GBs to the overall mass transport. Hence, the earlier work should be considered qualitative/heuristic/preliminary and applied to a different (albeit related) problem.

In the present work we overcome this deficiency of Ref. [23] by focusing on microstructural features that only exist in two-dimensions – small, “sinking” grains that are remote from neighboring GBs. Such “sinking” grains were not discussed in Ref. [23]. Here, our mass balance analysis for the neighborhood of the “sinking” grains accounts for all material that diffused out and the axisymmetric geometry implied by the microstructure is an explicit feature of the current manuscript. This analysis is key since it is what allowed us to extract Ni interface diffusion coefficients from the current experiments – this was not and could not be done based on the Amram *et al.*⁹ approach (Ref. [23] in the original manuscript).

References:

1. Park, J.-W. & Altstetter, C. J. The diffusion and solubility of oxygen in solid nickel. *Metall. Trans. A* **18**, 43–50 (1987).
2. Duffy, D. M., Harding, J. H. & Stoneham, A. M. The energies of point defects near metal/oxide interfaces. *J. Appl. Phys.* **76**, 2791–2798 (1994).
3. Tamura, T., Ishibashi, S., Terakura, K. & Weng, H. First-principles study of the rectifying properties of Pt/TiO₂ interface. *Phys. Rev. B* **80**, (2009).

4. Cho, E. *et al.* Segregation of oxygen vacancy at metal-HfO₂ interfaces. *Appl. Phys. Lett.* **92**, 233118 (2008).
5. Sharia, O., Tse, K., Robertson, J. & Demkov, A. A. Extended Frenkel pairs and band alignment at metal-oxide interfaces. *Phys. Rev. B* **79**, (2009).
6. Capron, N., Broqvist, P. & Pasquarello, A. Migration of oxygen vacancy in HfO₂ and across the HfO₂/SiO₂ interface: A first-principles investigation. *Appl. Phys. Lett.* **91**, 192905 (2007).
7. Yang, J., Youssef, M. & Yildiz, B. Predicting point defect equilibria across oxide hetero-interfaces: model system of ZrO₂/Cr₂O₃. *Phys. Chem. Chem. Phys.* **19**, 3869–3883 (2017).
8. Legros, M., Dehm, G., Arzt, E. & Balk, T. J. Observation of Giant Diffusivity Along Dislocation Cores. *Science* **319**, 1646–1649 (2008).
9. Amram, D., Klinger, L., Gazit, N., Gluska, H. & Rabkin, E. Grain boundary grooving in thin films revisited: The role of interface diffusion. *Acta Mater.* **69**, 386–396 (2014).

REVIEWERS' COMMENTS:

Reviewer #1 (Remarks to the Author):

In my opinion, the authors have addressed correctly the concerns of all the reviewers and modified the manuscript accordingly. I recommend its publication.

Reviewer #2 (Remarks to the Author):

The authors have adequately addressed all questions and comments which i had made in my Review. As such the manuscript can be accepted in the revised Version.

Reviewer #3 (Remarks to the Author):

All the points I raised in the first round of review are addressed and the manuscript is well revised and polished up to avoid unnecessary misunderstanding of any field of sub-fields based on which this study is performed. Furthermore, all the other points I had been concerned at the first round, more specifics about diffusion at various interfaces to account for mass balance before arriving at the model and the title whether anomalous or not, had been pointed out by other reviewers and the manuscript has been already revised accordingly. Thus, I find nothing to point out to the well polished manuscript that eventually leads readers to a simple-to-apply model which I believe is a core of this study, and I am happy to see the manuscript being as it deserves. Therefore, I suggest the editor to accept this manuscript for publication in the current form.

One apology: Needless to say, "heterogeneous grain boundaries" was a typo and it should have been "heterogeneous interfaces", though it seems the author fully understand what I tried to mean.